# Larval Aggregation of *Heortia vitessoides* Moore (Lepidoptera: Crambidae) and Evidence of Horizontal Transfer of Avermectin

**Shiping Liang [1], Jiacheng Cai [2], Xuan Chen [3], Zhengya Jin [1], Jinkun Zhang [1], Zhijia Huang [1], Liping Tang [1], Zhaohui Sun [1], Xiujun Wen [1] and Cai Wang [1,*]** 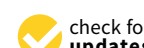

1   Guangdong Key Laboratory for Innovation Development and Utilization of Forest Plant Germplasm, College of Forestry and Landscape Architecture, South China Agricultural University, Guangzhou 510642, China; liangshiping@stu.scau.edu.cn (S.L.); jinzhengya@stu.scau.edu.cn (Z.J.); zhangjinkun@stu.scau.edu.cn (J.Z.); 20182031001@stu.scau.edu.cn (Z.H.); 20173076027@stu.scau.edu.cn (L.T.); sunzhaohui@scau.edu.cn (Z.S.); wenxiujun@scau.edu.cn (X.W.)
2   Department of Mathematics and Computer Science, Salisbury University, Salisbury, MD 21801, USA; JXCAI@salisbury.edu
3   Department of Biology, Salisbury University, Salisbury, MD 21801, USA; XXCHEN@salisbury.edu
*   Correspondence: wangcai@scau.edu.cn; Tel.: +86-020-85280256

**Abstract:** *Aquilaria sinensis* (Lour.) Gilg is an economically important tree species that produce the highly prized agarwood. In recent years, agarwood production has been seriously threatened by the outbreak of *Heortia vitessoides* Moore, a leaf-eating pest that shows gregariousness during the larval stage. However, little attention has been paid to the aggregation behavior of *H. vitessoides* larvae. In the present study, we collected 102 cohorts of *H. vitessoides* larvae (13,173 individuals in total) in the wild; 54 cohorts were comprised of the same-instar larvae, and 48 cohorts were comprised of larvae with different developmental stages (instars). In general, young larvae (<third instar) tended to form large aggregations, whereas older-instar larvae were either solitary or formed small aggregations. Laboratory studies showed a strong aggregation tendency in the newly hatched and second-instar larvae of *H. vitessoides*, whenever the individuals originated from the same or different sibling cohorts. In addition, all newly hatched larvae died within two days after they were isolated. When newly hatched larvae were initially assigned in 10-larvae cohorts (containing sibling individuals) or 20-larvae cohorts (either containing individuals originating from the same or different sibling cohorts), their larval survivorship, duration of larval stage, and adult emergence were not significantly different. Interestingly, combining avermectin-treated larvae (donors) with untreated ones (receptors) significantly decreased larval survivorship and adult emergence of receptors, indicating a horizontal transfer of avermectin among *H. vitessoides* larvae. This study enhances our understanding of the population ecology of *H. vitessoides*, and may bring novel insights into the management strategies against this pest.

**Keywords:** *Aquilaria sinensis* (Lour.) Gilg; *Heortia vitessoides* Moore; gregarious larvae; horizontal transfer; avermectin

## 1. Introduction

*Aquilaria sinensis* (Lour.) Gilg is a tree species distributed in southern China and Southeast Asia. This tree is one of the main sources of the fragrant product "agarwood", which has been widely used in traditional medicines, fragrance industries, and religious ceremonies. As a result of the scarcity and high demand of agarwood, the natural sources of *A. sinensis* trees have been seriously damaged

by illegal logging and trade. Nowadays, *A. sinensis* and all other species in the genus *Aquilaria* are classified as "Endangered" in Appendix II of the Convention on International Trade in Endangered Species of Fauna and Flora [1], and the international trade of wild-originated agarwood is strictly monitored and limited. To increase agarwood supply, *A. sinensis* trees have been planted on large scales in many provinces (e.g., Guangdong, Guangxi, Hainan, and Fujian) in southern China. Meanwhile, many efforts have been made to develop artificial agarwood-producing methods based on physical wounding of the trunks or injecting agarwood-induced chemicals into the xylem [2–5].

The cultivation of *A. sinensis* trees and modern agarwood-inducing techniques not only conserve wild *A. sinensis* resources but also increase the income of farmers [6]. However, the outbreak of a lepidopteran pest, *Heortia vitessoides* Moore (Lepidoptera: Crambidae), has been reported in many *A. sinensis* plantations, usually causing complete defoliation of the trees (Figure 1). Although the majority of *A. sinensis* trees can survive after *H. vitessoides* damage, they will be too weak for farmers to carry out agarwood-inducing operations [7]. In recent years, increasing attention has been paid on the biology of *H. vitessoides* [8–16]. One interesting behavior of *H. vitessoides* larvae is that they often form large aggregations to cooperatively feed on the leaves [10]. However, many aspects of the aggregation behavior of *H. vitessoides* larvae remain unknown. In the present study, we conducted field and laboratory studies to investigate: (1) the aggregation patterns of *H. vitessoides* larvae under field and laboratory conditions; (2) the factors that affect the aggregation behavior of *H. vitessoides* larvae; and (3) the biological significance of aggregation behavior on the survival and development of *H. vitessoides* larvae.

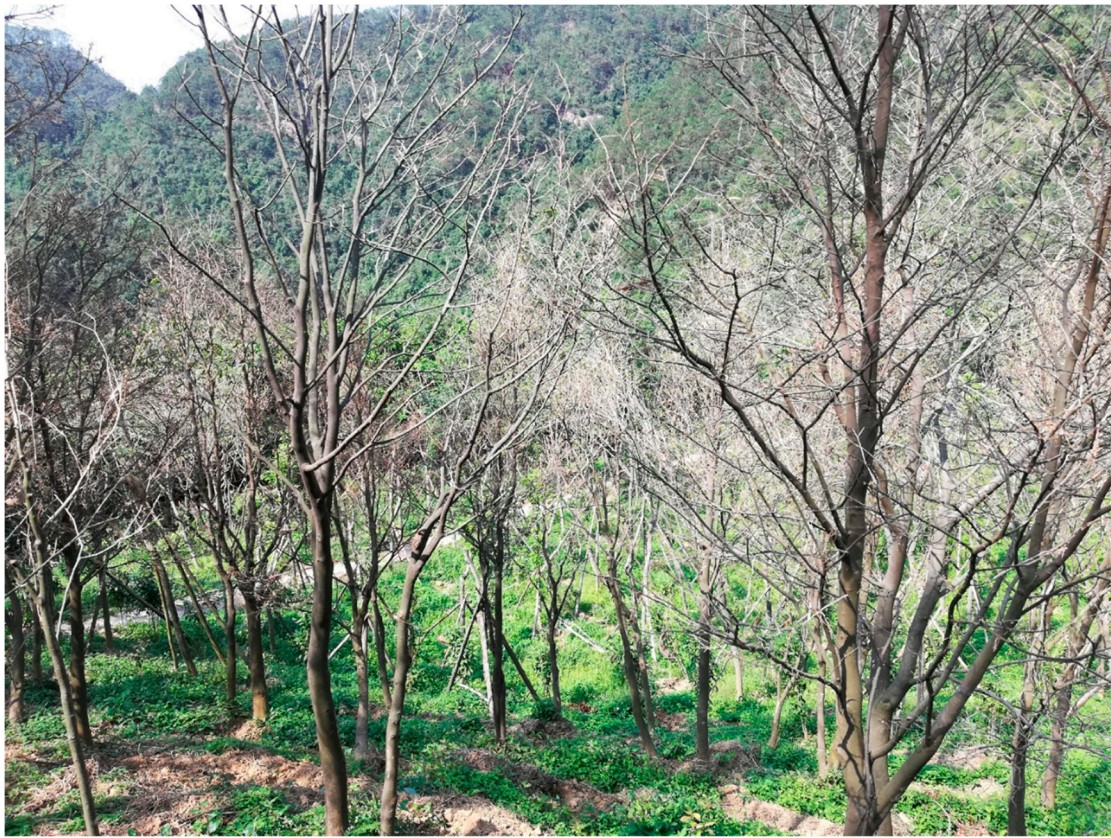

**Figure 1.** The outbreak of *Heortia vitessoides* Moore usually causes the complete defoliation of *Aquilaria sinensis* (Lour.) Gilg trees. The picture was taken in an *Aquilaria sinensis* plantation in Zhaoqin, Guangdong, China, on 13 November 2018 by Liping Tang.

Many chemical pesticides such as avermectin, spinosad, and trichlorfon have been applied to control *H. vitessoides* in the field [17], which may cause pesticide residue in the agarwood. Therefore, it

is important to reduce the use of pesticides. The horizontal transfer of chemical pesticides is a mode of action that requires a limited amount of pesticides, which can be transferred from pesticide-exposed (treated) individuals to unexposed (untreated) individuals. This method has been widely applied to control the gregarious insects such as termites [18,19], ants [20,21], and bed bugs [22]. Here we hypothesized that the chemical pesticide (i.e., avermectin) can be horizontally transferred within the aggregations of *H. vitessoides* larvae, and avermectin-transfer bioassays were conducted to verify this under laboratory conditions.

## 2. Materials and Methods

### 2.1. Field Study

The egg clusters and larval cohorts of *H. vitessoides* were collected from Tianlu Lake Park (113°8′–113°11′ E, 23°12′–23°14′ N), Guangzhou, China, from 15 October to 3 December, 2018. The damaged branches of *A. sinensis* trees were searched and cut using hand-operated sheers or clippers (an averruncator). A cohort of *H. vitessoides* larvae was defined as >3 larvae (commonly with close body contact) found on the same leaf (Figure 2A), or on several adjacent leaves (Figure 2B). In the latter case, large aggregations with a few sub-cohorts on different leaves can be observed (Figure 2B). Each larval cohort was placed into a 300 mL plastic container (upper side: 8.6 cm in diameter; bottom side: 6.6 cm in diameter; height: 6.2 cm). These larvae were brought to the laboratory within 3 h, and then preserved in 75% alcohol. The developmental stages (instars) of larvae in each cohort were determined using the methods provided by Qiao et al. [23], and counted. Egg clusters were also brought to the laboratory, and a high-resolution picture was taken of each cluster for egg counting.

### 2.2. Laboratory Study

#### 2.2.1. Insects

Each egg cluster collected in the field (as mentioned earlier) was placed in a Petri dish (9 cm in diameter) with fresh *A. sinensis* leaves and sealed to prevent dehydration. The hatched larvae in each Petri dish (considered as a sibling cohort because all individuals originated from the same egg cluster) were then transferred to a 300 mL plastic container. These larvae (newly hatched to second-instar) were used to set up laboratory experiments 1 and 2 (see below). Because *H. vitessoides* larvae from non-sibling cohorts showed a strong aggregation tendency (see results), we combined and reared third-instar larvae from different sibling cohorts together in 750 mL plastic containers (upper side: 14.1 cm in diameter; bottom side: 11.3 cm in diameter; height: 6.5 cm) to set up laboratory experiment 3. Fresh *A. sinensis* leaves were added to the containers each day, and any feces and leaf debris were removed regularly. Our previous studies indicated that *A. sinensis* leaves collected from different trees may affect the feeding preference of *H. vitessoides* larvae [10]. As a result, the larvae used in each experiment were fed on leaves collected from a single *A. sinensis* tree. All eggs and larvae were maintained in an environmental chamber setting at 12:12 light: dark schedule and 25 ± 1 °C.

#### 2.2.2. Experiment 1: Aggregation Tendency of *H. vitessoides* Larvae

Methods provided by Boulay et al. [24] were modified to investigate the aggregation tendency of *H. vitessoides* larvae that originated from the same or different sibling cohorts. The bioassay arenas were 2000 mL plastic containers (upper side: 24.8 cm in diameter; bottom side: 20.8 cm in diameter; height: 6.3 cm). Twenty *A. sinensis* leaves were pasted onto the bottom of each container using double-faced adhesive tape (Deli®, Ningbo, China), with each leaf contacted with the adjacent ones (Figure 2C). Newly hatched or second-instar larvae from six sibling cohorts were used in this experiment. Two treatments were set for the larvae of each instar: (1) 20 larvae from each sibling cohort were released into the container (one larva was randomly placed on the center of each leaf); and (2) 10 larvae from one sibling cohort and 10 from another (two cohorts were randomly selected from the 6 sibling cohorts)

were randomly released onto leaves. Each treatment was repeated 6 times. The bioassay arenas were kept at room temperature (24–26 °C), and the number of larvae aggregated on each leaf was recorded at 1, 3, 5, 24, and 48 h after release. The aggregation index (I) was calculated using the formula provided by Boulay et al. [24] as follows:

$$\text{Aggregation index (I)} = (SD)^2/\text{mean number of larvae in each leaf}$$

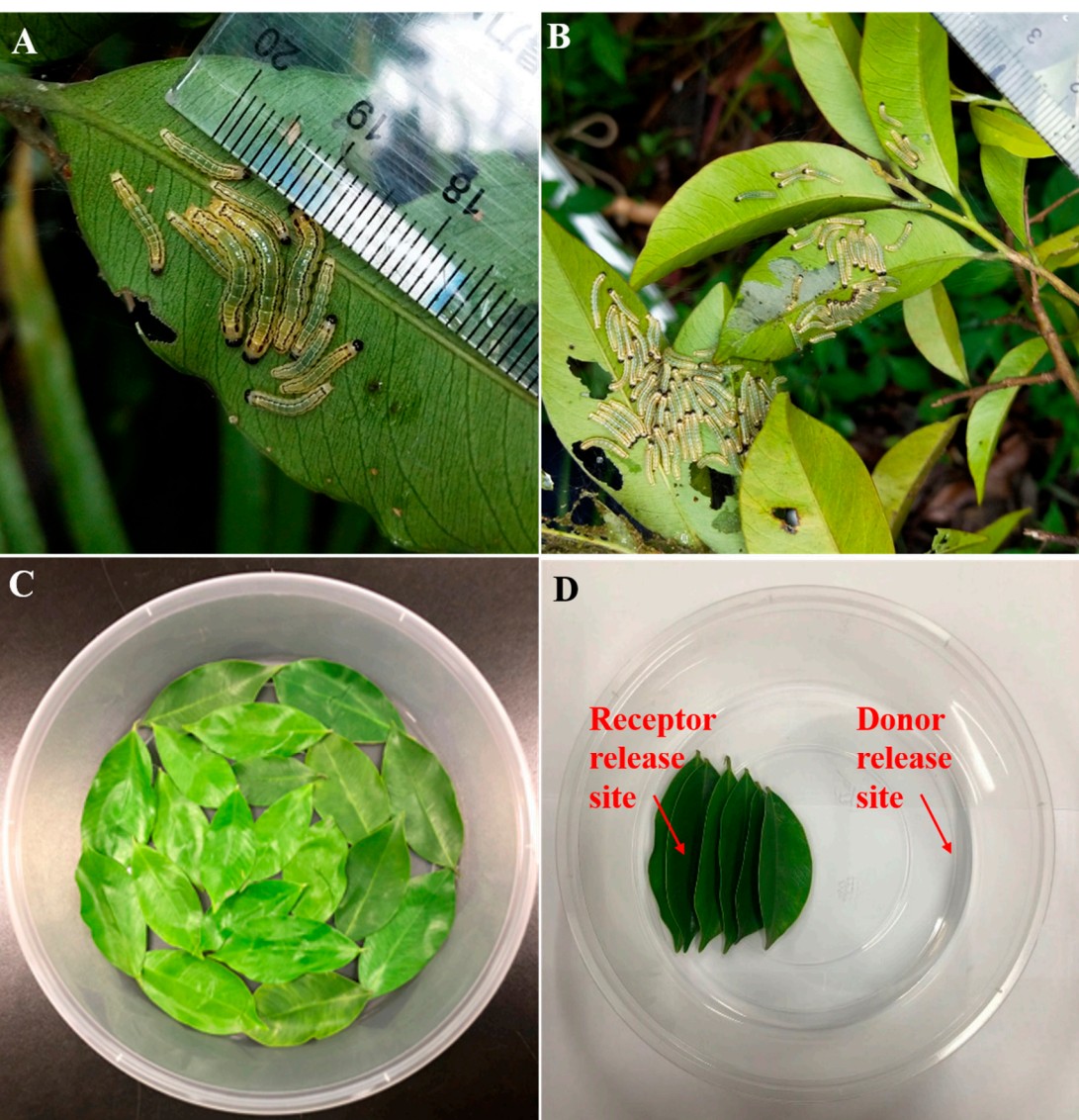

**Figure 2.** The aggregation behaviors of *Heortia vitessoides* larvae were investigated under field and laboratory conditions. In the field, a cohort of *Heortia vitessoides* larvae was defined as >3 larvae (commonly with close body contact) found on the same leaf (**A**), or on the several adjacent leaves (**B**). To investigate the aggregation tendency of *Heortia vitessoides* larvae, 20 *Aquilaria sinensis* leaves were pasted onto the bottom of a container using the double-faced adhesive tape, and a newly hatched or second-instar larva was placed onto each leaf (**C**). To evaluate the horizontal transfer of avermectin among gregarious larvae of *Heortia vitessoides*, 10 untreated larvae (receptors) were released onto one side of the container with six fresh *Aquilaria sinensis* leaves, and 10 donor larvae (previously released onto the avermectin-treated surfaces for 30 min) were then placed on the other side of the container (**D**). The first two pictures were taken by Zhijia Huang and the second two by Shiping Liang.

2.2.3. Experiment 2: Effects of Aggregation on *H. vitessoides* Survivorship and Development

The survivorship and development of *H. vitessoides* larvae were detected throughout the larval and early adult stages when they were assigned to different group sizes (1, 10, 20 larvae) and sibling status (originating from the same or different sibling cohorts) just after hatching. Eight sibling cohorts of newly hatched larvae were used in this experiment. The bioassays included 4 treatments: (1) single larva from each sibling cohort was transferred to a 300 mL plastic container with fresh *A. sinensis* leaves; (2) 10 larvae were randomly selected from each sibling cohort and transferred to a container; (3) 20 larvae were randomly selected from each sibling cohort and then released into a container; and (4) 10 larvae from one sibling cohort and 10 from another (two cohorts were randomly selected from the 8 sibling cohorts) were released into a container. Each treatment was repeated 8 times. Fresh *A. sinensis* leaves were added to the containers, and any feces, dead larvae, and leaf debris were removed each day. On day 11, larvae in each container were transferred to a bigger container (750 mL) because more space and food were needed as a result of the growth of the larvae. On day 18, wet sand (12% moisture) was added at the bottom of each 750 mL plastic container to a depth of 3 cm for the pupation of mature larvae. The bioassays were maintained in an environmental chamber (12:12 light–dark schedule and 25 ± 1 °C) throughout the experiment. The survivorship and developmental stages (instars) of larvae in each container were recorded daily until some individuals began to burrow into the sand for pupating after day 18. The duration of the larval stage (between hatching and burrowing into the sand for pupation) of each larva in each replication was recorded. We also recorded the number of emerged adults each day after pupating (until no more adult emerged for a 15-day period), and the emergence success was calculated as follows: emergence success (%) = (number of emerged adults/number of larvae initially released) × 100%.

2.2.4. Experiment 3: Horizontal Transfer of Avermectin among *H. vitessoides* Larvae

This experiment was conducted to investigate whether the chemical pesticide would transfer among gregarious larvae of *H. vitessoides* (transfer from treated larvae to untreated ones). Different concentrations of avermectin (Dugao®, Shijiazhuang, Hebei, China) solutions (0 (control), 100, or 200 ppm) were prepared. Each solution was sprayed once on the inner surfaces of a Petri dish and lid using a sprayer (Zhenxing Industrial Co., Ltd., Guangzhou, China)(0.92 ± 0.01 mL (mean ± SE, *n* = 10) solution was sprayed on the Petri dish or lid). After the solutions were air-dried (~40 min), 80 third-instar larvae were randomly selected and marked with a red ink spot on the dorsum, and then released onto the bottom of each Petri dish and covered. These larvae (donors) were allowed to move on the avermectin-treated surfaces for 30 min, and then transferred to an untreated Petri dish. The bioassay arenas were 1280 mL plastic containers (upper side: 19.1 cm in diameter; bottom side: 16.2 cm in diameter; height: 4.5 cm) with six fresh *A. sinensis* leaves placed on one side of the container (Figure 2D). Ten untreated larvae (receptors) were randomly selected and released onto leaves and allowed to acclimate for 1 h. Ten donors were then placed on the other side of the container (Figure 2D). Each of the three treatments (donor previously exposed to the Petri dish sprayed with 0, 100, or 200 ppm avermectin solutions) was repeated 7 times. The bioassays were prepared and maintained in an environmental chamber setting at a 12:12 light–dark schedule and 25 ± 1 °C.

After 12 h, donors and receptors in each container were separated and each group was transferred to a 450 mL plastic container (upper side: 11.2 cm in diameter; bottom side: 7.0 cm in diameter; height: 4.4 cm) with fresh *A. sinensis* leaves. Here we separated donors and receptors because the red ink spot on the donors would disappear within 1–2 days due to molting, and therefore we would not be able to distinguish donors and receptors. Larvae were reared as mentioned earlier. The survivorship of donors and receptors was recorded every 24 h after initial combination. On day 7, living donors and receptors were transferred to 750 mL plastic containers with fresh *A. sinensis* leaves and wet sand (12% moisture) on the bottom (depth = 3 cm). Because mature larvae began to burrow into the sand for pupating, we did not record larval survivorship after 7 days. The number of emerged donors and

receptors were recorded until no new adults were observed for 15 days, and emergence success was calculated as mentioned above.

### 2.3. Data Analysis

For the field study, one-way ANOVA (Proc Mixed, SAS 9.4, Cary, NC, USA) was conducted to compare the number of individuals within egg clusters and the same-instar cohorts (comprised of newly hatched, first-, second-, or third–fifth instar larvae) as well as different-instar mixed cohorts. For experiment 1 (the laboratory study), the aggregation tendency was studied by comparing the aggregation index of newly hatched or second-instar larvae using the mixed ANOVA (IBM SPSS Statistics 24, Chicago, IL, USA), with time as the within-subjects factor and sibling status as the between-subjects factor. To investigate the change of aggregation index through time, we performed one-way repeated measures ANOVA (IBM SPSS Statistics 24) for each sibling status with time as the within-subjects factor. For experiment 2, because all isolated larva died within two days (see results), we only compared the survivorship of 10-sibling larvae, 20-sibling larvae, and 20-non-sibling larvae using mixed ANOVA, with time as the within-subjects factor and treatment as the between-subjects factor. For experiment 3, we compared survivorship of donors or receptors using the mixed ANOVA (IBM SPSS Statistics 24), with time as the within-subjects factor and treatment as the between-subjects factor. For both experiment 2 and 3, the one-way ANOVAs (IBM SPSS Statistics 24) were then used to compare larval survivorship among treatments within each time point. The duration of the larval stage (experiment 2) and emergence success of adults (experiment 2 and 3) were compared between treatments using the one-way ANOVAs (Proc Mixed, SAS 9.4). The significance level was determined at $\alpha = 0.05$ for all tests.

## 3. Results

### 3.1. Field Study

In total, 58 egg clusters and 102 larval cohorts were obtained from the field (Table 1). Fifty-four cohorts were composed of larvae with the same developmental stage (same-instar cohorts), while 48 cohorts were composed of larvae with different instars (different-instar mixed cohorts). The mean number of individuals in the same-instar cohorts containing the first-instar larvae was significantly higher than the mean number of eggs in the egg clusters, and both were not significantly different from the mean number of individuals in the same-instar cohorts containing newly hatched larvae (Figure 3). The number of individuals in the same-instar cohorts containing third-, fourth-, or fifth-instar larvae and different-instar mixed cohorts was similar, and all were significantly lower than that in the same-instar cohorts containing newly hatched or first-instar larvae ($F = 14.84$, $df = 5, 154$, $p < 0.0001$; Figure 3). The number of larvae in different-instar mixed cohorts was quite variable, ranging from several to hundreds of individuals (Table 2). Interestingly, 70.8%, 77.1%, and 52.1% of different-instar mixed cohorts contained second-, third-, and fourth-instar larvae, respectively, whereas only a few different-instar mixed cohorts contained newly hatched and first-instar larvae (Table 2).

**Table 1.** Number of egg clusters and larval cohorts collected in each collection date.

| Collection Date | Egg Clusters | Same-Instar Cohorts | Different-Instar Mixed Cohorts |
|---|---|---|---|
| 15 October 2018 | 1 | 26 | 38 |
| 29 October 2018 | 0 | 4 | 7 |
| 21 November 2018 | 2 | 0 | 0 |
| 24 November 2018 | 2 | 0 | 0 |
| 27 November 2018 | 42 | 7 | 0 |
| 3 December 2018 | 11 | 17 | 3 |
| Total | 58 | 54 | 48 |

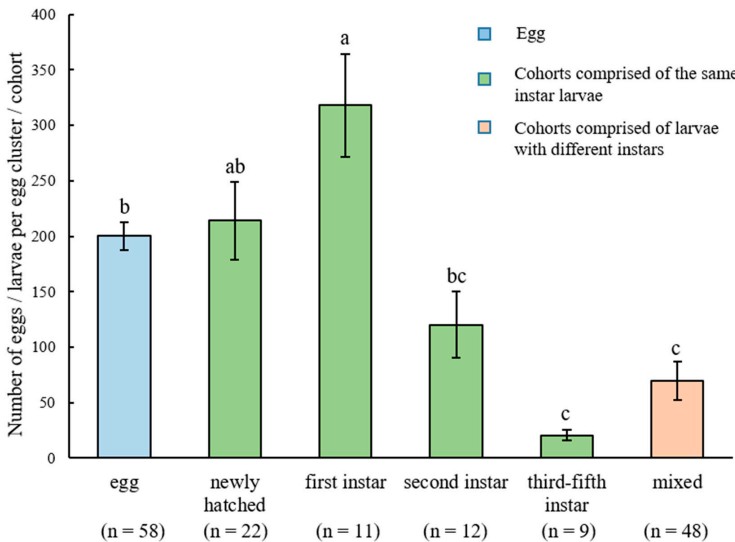

**Figure 3.** The number of *Heortia vitessoides* individuals (mean ± SE) in egg clusters and in the cohorts containing the same-instar or different-instar larvae. Different letters indicate significant differences ($p < 0.05$).

**Table 2.** The number of larvae in each different-instar mixed cohort.

| Cohort | Newly Hatched Larvae | First-Instar Larvae | Second-Instar Larvae | Third-Instar Larvae | Fourth-Instar Larvae | Fifth-Instar Larvae | Total |
|---|---|---|---|---|---|---|---|
| 1 | 411 | 276 | | | | | 687 |
| 2 | 174 | 161 | | | | | 335 |
| 3 | | | 68 | 216 | | | 284 |
| 4 | | 20 | 224 | 8 | | | 252 |
| 5 | | | 27 | 171 | | | 198 |
| 6 | | 59 | 82 | | | | 141 |
| 7 | | 140 | | 1 | | | 141 |
| 8 | | 72 | 58 | | | | 130 |
| 9 | | 52 | 77 | | | | 129 |
| 10 | | 2 | 118 | | | | 120 |
| 11 | | | 83 | 28 | | | 111 |
| 12 | | | 68 | 1 | | | 69 |
| 13 | | | 5 | 50 | | | 55 |
| 14 | | | 5 | 31 | 12 | | 48 |
| 15 | | | 47 | 1 | | | 48 |
| 16 | | | 4 | 42 | | | 46 |
| 17 | | | 4 | 39 | 1 | | 44 |
| 18 | | | 6 | 19 | 13 | | 38 |
| 19 | | | 1 | 23 | 5 | | 29 |
| 20 | | | 8 | 7 | 14 | | 29 |
| 21 | | | 6 | 21 | | | 27 |
| 22 | | | 13 | 13 | | | 26 |
| 23 | | 2 | 23 | | | | 25 |
| 24 | | | 2 | 17 | 2 | | 21 |
| 25 | | | 3 | 17 | | | 20 |
| 26 | | 2 | 5 | 10 | 2 | | 19 |
| 27 | | | 1 | 15 | 3 | | 19 |
| 28 | | | | 13 | 6 | | 19 |
| 29 | | | 2 | 10 | 6 | | 18 |
| 30 | | | | 12 | 6 | | 18 |
| 31 | | | 2 | 6 | 7 | 1 | 16 |
| 32 | | 2 | 6 | 7 | 1 | | 16 |
| 33 | | | 6 | 8 | 1 | | 15 |
| 34 | | | 6 | 8 | 1 | | 15 |
| 35 | | | | 8 | 6 | | 14 |
| 36 | | | | 2 | | 12 | 14 |
| 37 | | | | 10 | 3 | | 13 |
| 38 | | | | 1 | 4 | 7 | 12 |
| 39 | | | 3 | 8 | | | 11 |
| 40 | | | 8 | 3 | | | 11 |
| 41 | | | 3 | 6 | 2 | | 11 |
| 42 | | | 2 | 3 | 4 | | 9 |
| 43 | | | 2 | | 7 | | 9 |
| 44 | | | | 3 | | 6 | 9 |
| 45 | | | | | 4 | 3 | 7 |
| 46 | | | | 1 | 5 | | 6 |
| 47 | | | | | 4 | 1 | 5 |
| 48 | | | | | 2 | 1 | 3 |

## 3.2. Laboratory Study

### 3.2.1. Experiment 1: Aggregation Tendency of *H. vitessoides* Larvae

The *H. vitessoides larvae* showed a strong aggregation tendency, regardless of developmental stages (newly hatched or second-instar) and sibling status (originated from the same or different sibling cohorts) (Figure 4). The effect of sibling status (newly hatched larvae: $F = 0.353$, $df = 1, 10$, $p = 0.566$; second-instar larvae: $F = 2.210$, $df = 1, 10$, $p = 0.168$) and interaction effect between sibling status and time (newly hatched larvae: $F = 0.312$, $df = 1.904, 19.039$, $p = 0.725$; second-instar larvae: $F = 1.372$, $df = 3.020, 30.201$, $p = 0.270$) were not significant, but the aggregation index significantly increased with time for both newly hatched ($F = 34.443$, $df = 1.904, 19.039$, $p < 0.001$) and second-instar larvae ($F = 10.975$, $df = 3.020, 30.201$, $p < 0.001$; Figure 5; Supplementary Materials: Tables S1 and S2).

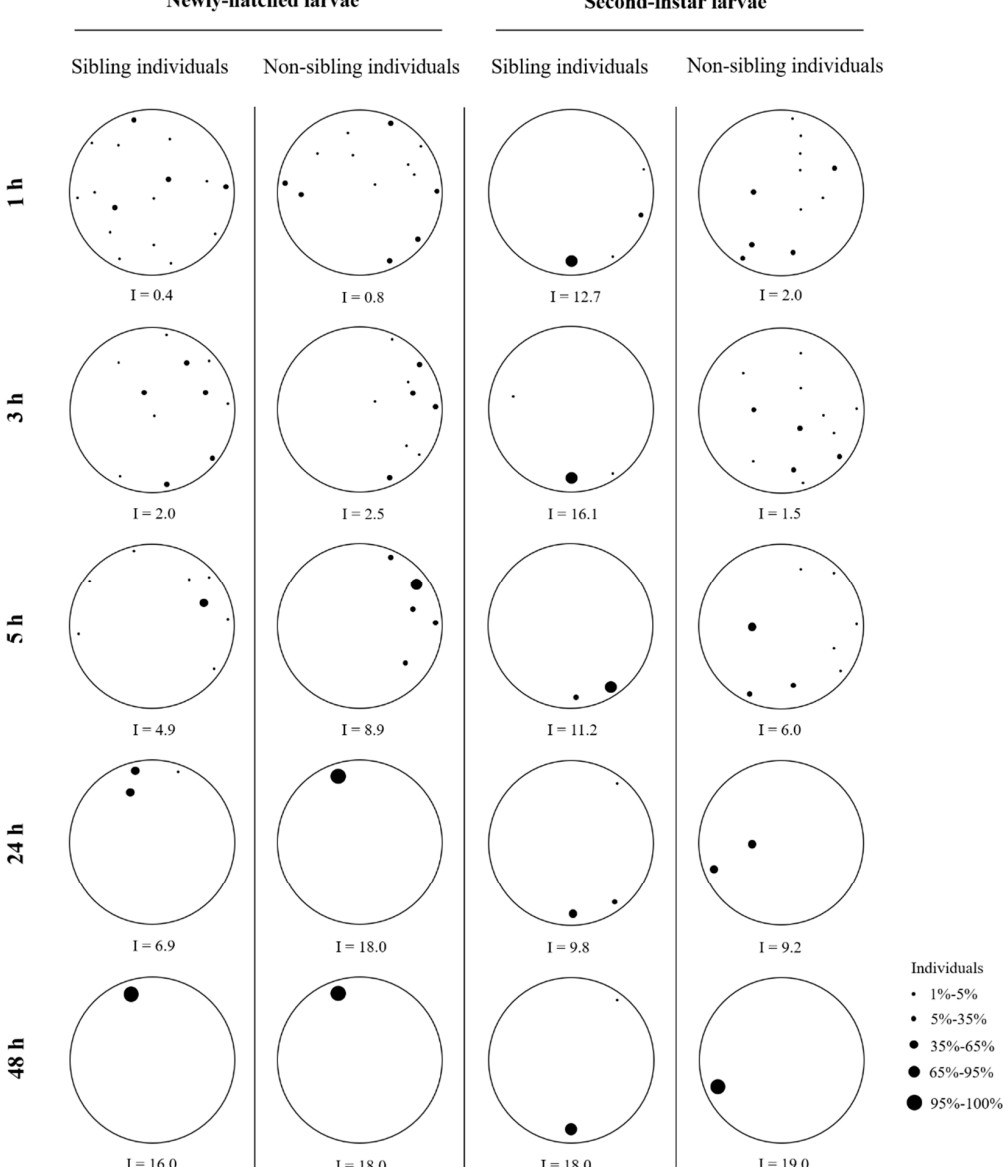

**Figure 4.** Examples of aggregation behavior of newly hatched or second-instar larvae of *Heortia vitessoides* in the arena containing sibling individuals (20 larvae originated from the same sibling cohort), or non-sibling individuals (10 larvae originated from one sibling cohort and 10 from another sibling cohort).

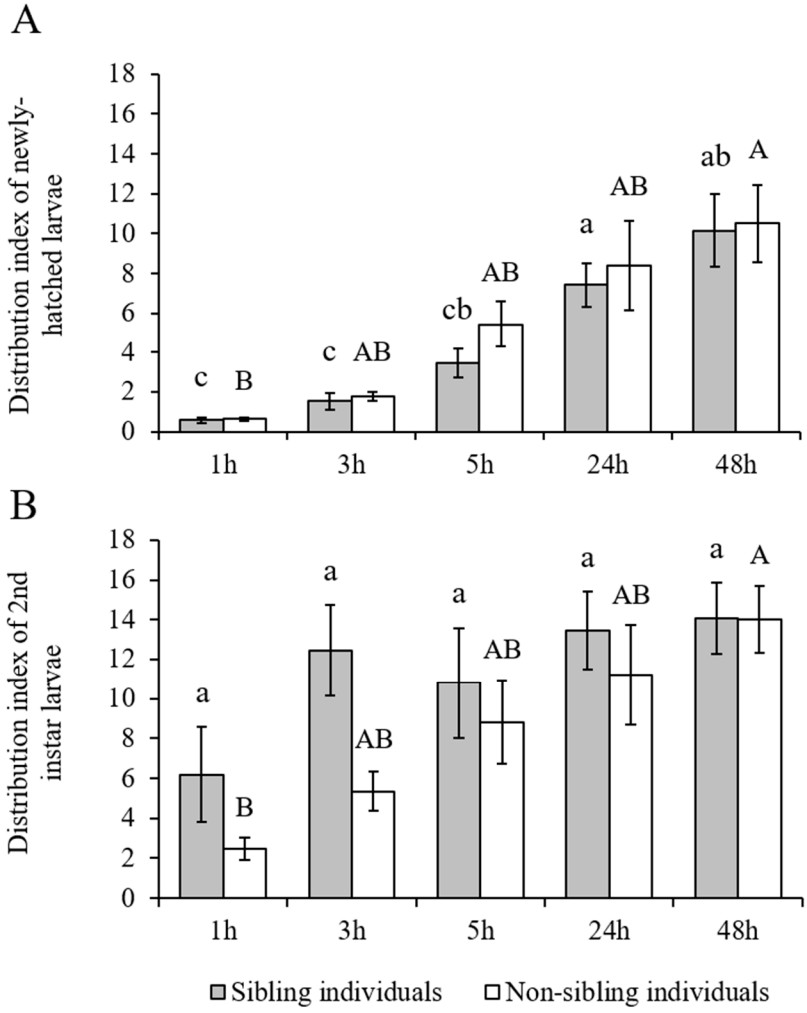

**Figure 5.** (**A**) Aggregation index of newly hatched or (**B**) second-instar larvae of *Heortia vitessoides* in the arena containing sibling individuals (20 larvae originated from the same sibling cohort), or non-sibling individuals (10 larvae originated from one sibling cohort and 10 from another sibling cohort). Different lower-case letters indicate significant differences in aggregation index of sibling individuals compared among the five time points ($p < 0.05$). Different capital letters indicate significant differences in aggregation index of non-sibling individuals compared among the five time points ($p < 0.05$).

3.2.2. Experiment 2: Effects of Aggregation on *H. vitessoides* Survivorship and Development

All newly hatched larvae died within two days after they were isolated (Figure 6A). No evidence of leaf consumption was observed before these larvae died. The effect of time ($F = 19.272$, $df = 2.560$, 46.080, $p < 0.001$) and the interaction effect of time and treatments ($F = 2.647$, $df = 5.120$, 46.080, $p = 0.034$) were significant, but there was no significant difference in larval survivorship between 10-larvae cohorts (containing sibling individuals) and 20-larvae cohorts (either containing individuals from the same or different sibling cohorts) ($F = 1.312$, $df = 2$, 18, $p = 0.294$; Figure 6A; Supplementary Material: Table S3). In addition, the duration of the larval stage ($F = 0.95$; $df = 2$, 21; $p = 0.4031$) and emergence success of adults ($F = 2.39$; $df = 2$, 291; $p = 0.0936$) were similar when compared between 10-larvae cohorts and 20-larvae cohorts (either containing individuals from the same or different sibling cohorts) (Figure 6B,C). Interestingly, the development of larvae in some replications was asynchronous (Supplementary Materials: Table S4), even for the larvae originating from the same sibling cohorts (hatched from the same egg cluster at the same time) (Table 3).

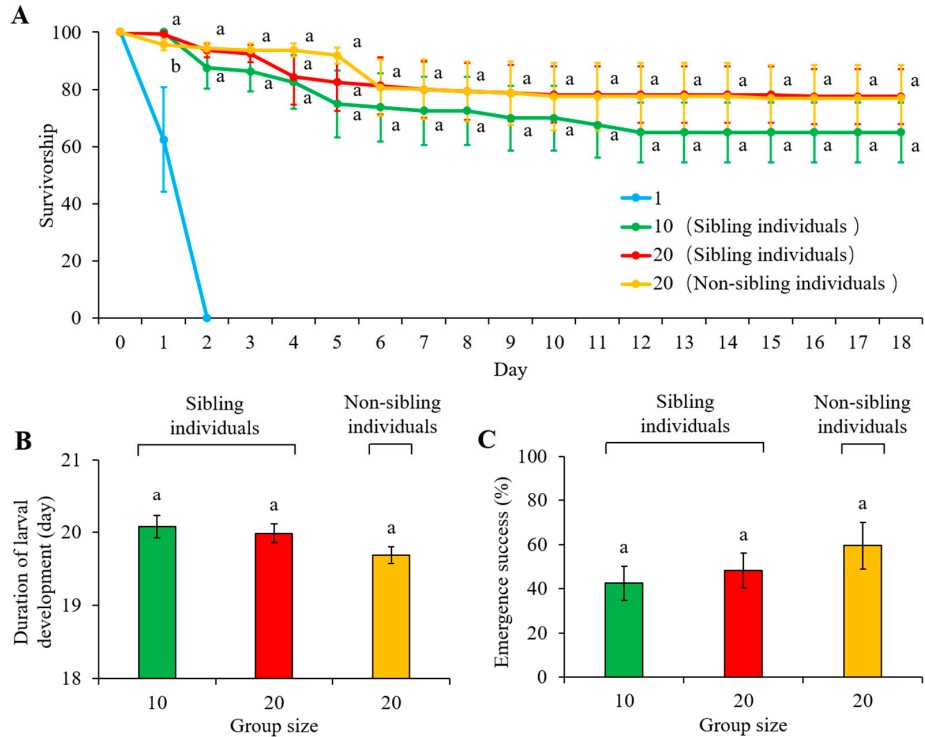

**Figure 6.** (**A**) Larval survivorship, (**B**) duration of larval stage, and (**C**) emergence success of adults of *Heortia vitessoides* individuals in the treatments containing: (1) single larva originated from each sibling cohort; (2) 10 larvae originated from the same sibling cohort; (3) 20 larvae originated from the same sibling cohort; and (4) 10 larvae originated from one sibling cohort and 10 from another. Since all newly hatched larvae died within two days after they were isolated, the comparisons were conducted for the latter three treatments. Data were presented as mean ± SE, and different letters indicate significant differences ($p < 0.05$).

**Table 3.** The developmental stages (I: first instar; II: second instar; III: third instar; IV: fourth instar; V: fifth instar; P: pupae) of *Heortia vitessoides* individuals during the whole larval stage after the newly hatched larvae were initially assigned into 20-larvae cohorts (containing sibling individuals). Note that the larvae of different developmental stages (instars) can be observed within the same sibling cohort at the same period.

| Day | Replication | | | | | | | |
|-----|---|---|---|---|---|---|---|---|
| | 1 | 2 | 3 | 4 | 5 | 6 | 7 | 8 |
| 2 | 19 I | 18 I | 18 I | 19 I | 16 I | 20 I | 20 I | 20 I |
| 4 | 19 II | 18 II | 15 II | 19 II | 4 I | 20 II | 20 II | 20 I |
| 6 | 18 II | 17 II | 14 II | 19 II | 3 II | 19 II | 18 II, 2 III | 20 II |
| 8 | 18 III | 17 III | 13 III | 18 III | 3 II | 18 III | 3 II, 17 III | 20 III |
| 10 | 18 IV | 6 IV, 10 III | 3 III, 10 IV | 17 IV | 3 III | 18 IV | 20 IV | 1 III, 19 IV |
| 12 | 18 IV | 16 IV | 3 III, 10 IV | 17 IV | 3 III | 18 IV | 20 IV | 20 IV |
| 14 | 18 IV | 16 IV | 2 III, 11 IV | 17 IV | 3 III | 18 V | 20 IV | 20 IV |
| 16 | 18 V | 16 V | 2 IV, 11 V | 17 V | 3 IV | 18 V | 19 V | 20 V |
| 18 | 18 V | 16 V | 13 V | 17 V | 3 V | 18 V | 19 V | 19 V |
| 20 | 3 V, 15 P | 16 V | 4 V, 9 P | 17 P | 3 V | 18 P | 4 V, 15 P | 5 V, 15 P |
| 22 | 3 V, 15 P | 16 V | 3 V, 10 P | 17 P | 3 V | 18 P | 4 V, 15 P | 5 V, 15 P |
| 24 | 18 P | 16 P | 3 V, 10 P | 17 P | 1 V, 1 P | 18 P | 19 P | 3 V, 17 P |
| 26 | 18 P | 16 P | 1 V, 11 P | 17 P | 1 V, 1 P | 18 P | 19 P | 1 V, 19 P |
| 28 | 18 P | 16 P | 11 P | 17 P | 2 P | 18 P | 19 P | 19 P |

### 3.2.3. Experiment 3: Horizontal Transfer of Avermectin among *H. vitessoides* Larvae

For larval survivorship of both donors and receptors, the effect of time (donors: $F = 273.676$, $df = 3.258, 58.652$, $p < 0.001$; receptors: $F = 16.452$, $df = 1.353, 24.349$, $p < 0.001$) and treatment (donors: $F = 332.512$, $df = 2, 18$, $p < 0.001$; receptors: $F = 5.664$, $df = 2, 18$, $p = 0.012$) and the interaction between

time and treatment (donors: *F* = 73.615, *df* = 6.517, 58.652, *p* < 0.001; receptors: *F* = 4.650, *df* = 2.705, 24.349, *p* = 0.012) were significant. Comparisons within each time point showed that the donors previously exposed to the Petri dish sprayed with 100 or 200 ppm avermectin solution had significantly lower survivorships compared with control donors (previously exposed to the Petri dish sprayed with distilled water) (Figure 7A, Supplementary Materials: Table S5). The receptors combined with donors previously exposed to the Petri dish sprayed with 100 ppm avermectin solution had significantly lower survivorship compared with control receptors after day 4 (Figure 7B, statistic results are shown in Supplementary Materials: Table S6). While larval survivorship of receptors combined with donors previously exposed to the Petri dish sprayed with 200 ppm avermectin solution was lower than the control on days 4 and 5; it did not significantly differ from either the control or 100 ppm avermectin treatments on days 6 and 7 (Figure 7B). The emergence success of receptors combined with donors previously exposed to the Petri dish sprayed with 100 ppm avermectin solution was significantly lower than that of the control receptors (*F* = 29.71; *df* =5, 36; *p* < 0.0001) (Figure 7C).

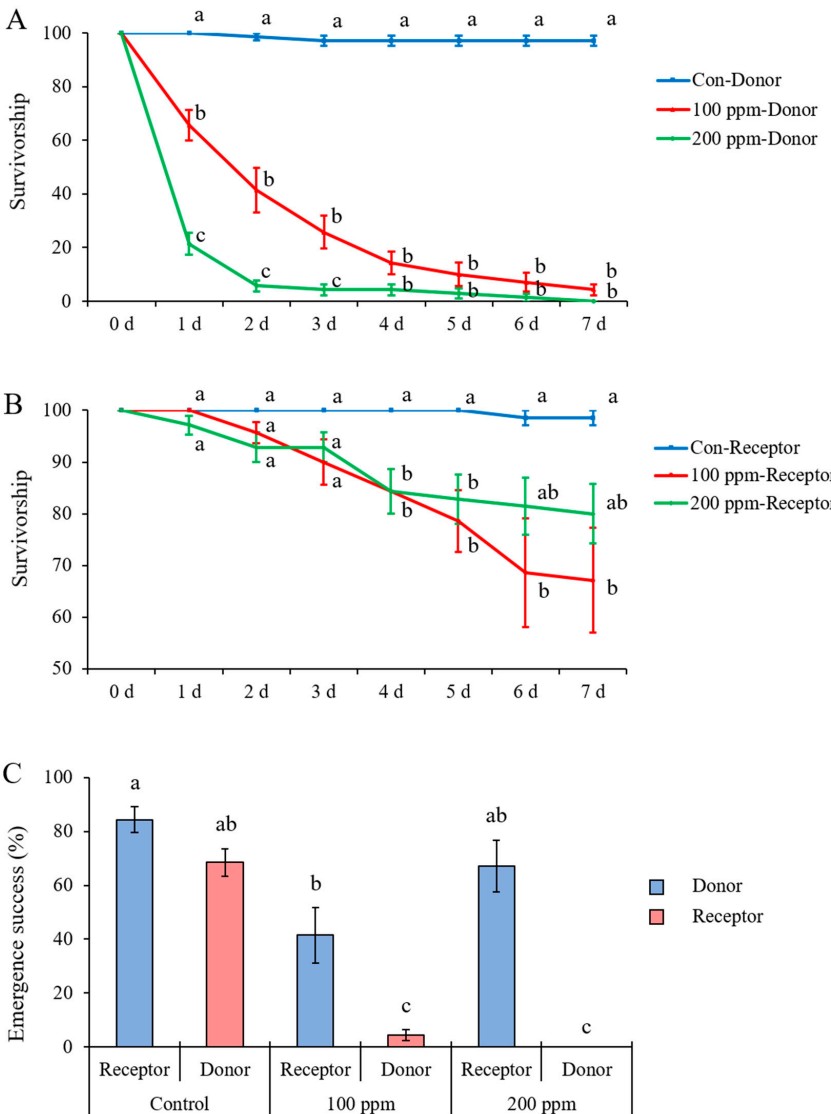

**Figure 7.** (**A**) Larval survivorship of donors (**B**) and receptors and (**C**) emergence success of adults of *Heortia vitessoides* individuals in each treatment which donors were previously exposed to the Petri dish sprayed with 0, 100, or 200 ppm avermectin solution. Data were presented as mean ± SE, and different letters indicate significant differences (*p* < 0.05).

## 4. Discussion

Compared to adults, animals in larval stages are vulnerable because of their limited mobility and foraging abilities. Social aggregation is one of the strategies that provide protective and facilitation effects aiding the survival and development of larvae [25], and has arisen independently in diverse animal taxa [26–30]. The majority of insects have immature stages, and the aggregation behaviors of larvae have been reported in many lineages including true flies [24,31–33], sawflies [34–37], and beetles [38–40].

Although many lepidopteran adults lay single eggs and their larvae are solitary [41], some species lay eggs in clusters and the larvae show gregarious behavior. For example, Stamp [42] estimated that 5%–15% of butterfly species have gregarious larval stages. Larvae of some serious agricultural and forestry lepidopteran pests, such as *Spodoptera litura* (Fabr.) (Lepidoptera: Noctuidae), *Malacosoma americanum* (Fabr.) (Lepidoptera: Lasiocampidae) and *Hemileuca Lucina* Hy. Edw. (Lepidoptera: Saturniidae), are also gregarious [43–45]. Most of these species show gregariousness during first third or fourth instars with the last one or two instars being solitary [46], and the number of individuals in each aggregation was usually low [47]. Our field study also showed that *H. vitessoides* larvae are gregarious during early instars, and later instar larvae are largely solitary, though some of them still formed small aggregations. Some *H. vitessoides* aggregations can be very large in the wild, usually containing hundreds of individuals (Figure 3), which may be a result of cohort merging because: (1) the mean number of individuals in the first-instar cohorts was significantly higher than that of the eggs in the clusters; and (2) larvae of different developmental stages were usually observed within the same cohort (though this also can be caused by the asynchronous development of larvae within the same cohorts, as shown in Table 3). We believe that the aggregation behaviors of *H. vitessoides* larvae are an important reason accounting for the severe defoliation caused by this pest. Since the early instar larvae are highly concentrated, we suggest artificially removing *H. vitessoides* aggregations from *A. sinensis* trees before the dispersing of later instar larvae.

The group-size effects on the survival and development of gregarious larvae and the related potential mechanisms have been investigated in many lepidopteran species. For example, Clark and Faeth [48] reported that the larvae of *Chlosyne lacinia* (Geyer) (Lepidoptera: Nymphalidae) developed faster when the group size was bigger, because individuals in larger groups found it easier to overcome the physical toughness of host plant leaves. Santana et al. [28] reported that isolated larva of *Ascia monuste* (Godart) (Lepidoptera: Pieridae) showed lower permanency on the host plant than that of aggregations, and large aggregations had a lower predation rate by natural enemies compared with small ones. In addition, Klok and Chown [49] reported that group living of second- and third-instar larvae of *Imbrasia belina* (Westwood) (Lepidoptera: Saturniidae) helped to maintain the body temperature of individuals due to "the accumulation of higher heat loads by the larger masses of large aggregations". Interestingly, Inouye and Johnson [41] found that the feeding rate of first-instar larvae of *Chlosyne poecile* (Felder) (Lepidoptera: Nymphalidae) significantly increased with group size, whereas the opposite trend was observed in second-instar larvae. Since large groups of second-instars larvae of *C. poecile* are commonly found in the field [41], benefits other than feeding simulation should exist. Similar to many previous studies, our study showed that isolated larva of *H. vitessoides* did not feed on leaves and eventually died within a short period, indicating that gregariousness may play a critical role in facilitating feeding for newly hatched larvae. However, we found that group size did not affect survivorship and development of *H. vitessoides* larvae under laboratory conditions (Figure 6). Future field studies are needed to investigate if the group size affects predation risks and thermoregulation of *H. vitessoides* larvae under natural conditions.

Many larval aggregations of lepidopteran species are composed of sibling individuals [48]. In a few species such as *Euselasia chrysippe* Bates (Lepidoptera: Riodinidae), non-sibling cohorts sometimes merge to form bigger aggregations, indicating that "benefits of living in large groups outweigh the costs of intra- or inter-specific competition in this species" [47]. Likewise, our studies showed that *H. vitessoides* larvae had a strong tendency to form the aggregations, whether they initially originated

from the same or different sibling cohorts. This result probably showed that the larval gregariousness of *H. vitessoides* was not kin-selected. The developmental stage is another factor that might affect larval aggregation. For example, Cornell et al. [44] reported that the first-instar larvae of *H. lucina* barely aggregated, whereas older instar larvae can quickly form aggregations. Although our study did not compare the aggregation index between newly hatched and second-instar larvae of *H. vitessoides*, larvae in both stages successfully formed the aggregations during the experiment (Figures 4 and 5).

Our study showed the horizontal transfer of avermectin among *H. vitessoides* larvae. To our best knowledge, this is the first report on pesticide horizontal transfer among gregarious larvae of a lepidopteran pest. To separate donors and receptors, previous studies on social insects, such as termites, commonly fed donors with dyes including Nile blue A and Sudan Red 7B [50–52]. However, marking the donors of *H. vitessoides* larvae would be challenging. Our preliminary studies showed that *H. vitessoides* larvae that fed on leaves treated with low concentrations of Nile blue A or Sudan Red 7B showed high mortality and limited mobility. Marking the body with inks did not affect moving and aggregation behaviors of *H. vitessoides* larvae; however, the ink spots easily faded after a short period. As a result, in the present study, donors were only allowed to combine with receptors for 12 h. The limited combining duration may to some extent reduce the effectiveness of avermectin transfer.

The horizontal transfer of pesticides among termites and ants can be caused by trophallaxis, grooming, and body contact among donors and receptors, as well as contact between receptors and pesticide-contaminated areas [53,54]. Since trophallaxis and grooming behaviors have not been observed in the gregarious larvae of *H. vitessoides*, contacting may account for the horizontal transfer of avermectin in this species. Here, donor larvae exposed to the Petri dish sprayed with the high concentration (200 ppm) of avermectin solution showed reduced movement and aggregation behaviors, which might reduce the chance of contact between donors and receptors. Therefore, donors treated with a high concentration of avermectin did not cause significantly lower larval survivorship and adult emergence of receptors as compared to the controls (Figure 7). This result showed that high concentrations of pesticide might negatively affect the effectiveness of horizontal transfer, and should be avoided in *H. vitessoides* management. It is important to note that this experiment is somewhat preliminary. Future studies are needed to investigate the factors (e.g., larval instars, donor–receptor ratio) that may influence horizontal transfer of various pesticides among gregarious larvae of *H. vitessoides.*

## 5. Conclusions

Our field study showed that *H. vitessoides* larvae are gregarious during early instars, and later instar larvae are largely solitary, though some of them still formed small aggregations. Some *H. vitessoides* aggregations can be very large in the wild, usually containing hundreds of individuals. In the laboratory, *H. vitessoides* larvae had a strong tendency to form the aggregations, whether they initially originated from the same or different sibling cohorts. When newly hatched larvae were isolated, they did not feed on leaves and eventually died within a short period. However, group size and sibling status did not affect survivorship and development of *H. vitessoides* larvae. In addition, our study is the first to show the horizontal transfer of avermectin among *H. vitessoides* larvae.

**Supplementary Materials:** The following are available online at http://www.mdpi.com/1999-4907/10/4/331/s1, Table S1: Statistics Analysis for Figure 5A, Table S2: Statistics Analysis for Figure 5B, Table S3: Statistics Analysis for Figure 6A, Table S4: The developmental stages (I: first instar; II: second instar; III: third instar; IV: fourth instar; V: fifth instar; P: pupae) of *Heortia vitessoides* individuals during the whole larval stage after the newly hatched larvae were initially assigned into 10 or 20-larvae cohorts, Table S5: Statistics Analysis for Figure 7A, Table S6: Statistics Analysis for Figure 7B.

**Author Contributions:** Formal Analysis, S.L., J.C., X.C. and C.W.; Investigation, S.L., Z.J., J.Z., Z.H., L.T., and C.W.; Resources, Z.S. and X.W.; Data Curation, S.L. and C.W.; Writing-Original Draft Preparation, S.L. and C.W.; Writing-Review & Editing, J.C., X.C., Z.J., J.Z., Z.H., L.T., Z.S. and X.W.; Visualization, S.L., J.C., X.C. and C.W.; Supervision, C.W.; Project Administration, C.W.; Funding Acquisition, C.W.

**Funding:** This research was funded by the Guangdong Natural Science Foundation (Grant No. 2016A030310445).

**Acknowledgments:** We sincerely thank Qinxi Xie, Hongpeng Xiong, and Wenquan Qin (College of Forestry and Landscape Architecture, South China Agricultural University) for valuable helps in *H. vitessoides* collection and rearing.

**Conflicts of Interest:** The authors declare no conflict of interest.

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
