# Peer review of "Larval Aggregation of Heortia vitessoides Moore (Lepidoptera: Crambidae) and Evidence of Horizontal Transfer of Avermectin"

_forests, doi:10.3390/f10040331_

Reviewer 1 Report

Review of Forests 479524

 This research focuses on the aggregation behavior of Heortia vitessoides and its impact on larval development and survival, as well as the horizontal transfer of insecticide among larvae. The methodology is solid and the results and discussion are thorough. The paper could benefit from some more English language editing but the content is scientifically rigorous.

 specific comments:

L 37: I suggest omitting “non-wood” because I think that it will confuse the reader. Instead you might revise like this:  “…produce the fragrant product “agarwood”, which is produced from the heartwood of trees infected by Phialophora parasitica and is widely used in traditional medicines, fragrance industries, and religious ceremonies.”

L109-111: Statement isn’t clear. Why did the previous observation that larvae preferred leaves collected from different trees, prompt you to use leaves from a single tree?

L196: It appears that you also did some post-hoc pairwise comparisons of treatments. Please include statistical methods for these.

L246-248: I would think that larvae from the same egg cluster are likely to have synchronous development, but this sentence seems to imply that this was an unexpected result. I wonder if this was not what you meant to imply.

L257: I think that Table 1 could be moved to supplemental information.

L272-275: It might be clearer to say something like:

While larval survivorship of receptors combined with donors previously 272 exposed to the Petri dish sprayed with 200 ppm avermectin solution was lower than the control on days 4 and 5, it did not significantly differ from either the control or 100 ppm avermectin treatments on days 6 and 7 (Fig. 7B).

L302-304: Do you mean: “this also can be caused by the asynchronous development of larvae within the same cohorts”?

L332: Replace “whenever” with “whether”.

Author Response

Dear Reviewer,

We sincerely thank you for valuable comments on our manuscript. We carefully went through your comments and revised the manuscript accordingly. We are resubmitting the revised manuscript. Following is the response to your comments. 

L 37: I suggest omitting “non-wood” because I think that it will confuse the reader. Instead you might revise like this:  “…produce the fragrant product “agarwood”, which is produced from the heartwood of trees infected by Phialophora parasitica and is widely used in traditional medicines, fragrance industries, and religious ceremonies.”

Response: We omitted “non-wood” as suggested by the reviewer.

L109-111: Statement isn’t clear. Why did the previous observation that larvae preferred leaves collected from different trees, prompt you to use leaves from a single tree?

Response: Our previous study showed that leaves collected from different trees may affect feeding behaviors of larvae. As a result, we used the leaves collected from a single tree to avoid any bias caused by the tree-effects. 

L196: It appears that you also did some post-hoc pairwise comparisons of treatments. Please include statistical methods for these.

Response: We thank for this comment by the reviewer. There are too many results of post-hoc pairwise comparisons, and we presented them in the supplemental materials.

L246-248: I would think that larvae from the same egg cluster are likely to have synchronous development, but this sentence seems to imply that this was an unexpected result. I wonder if this was not what you meant to imply.

Response: The development of larvae from the same egg cluster were asynchronous. We replaced “synchronous” with “asynchronous” in the revised manuscript.

L257: I think that Table 1 could be moved to supplemental information.

Response: Larvae from the same egg cluster had asynchronous development. We presented this unexpected result in the Table 1.

L272-275: It might be clearer to say something like:

While larval survivorship of receptors combined with donors previously 272 exposed to the Petri dish sprayed with 200 ppm avermectin solution was lower than the control on days 4 and 5, it did not significantly differ from either the control or 100 ppm avermectin treatments on days 6 and 7 (Fig. 7B).

Response: We revised this sentence as suggested by the reviewer.

L302-304: Do you mean: “this also can be caused by the asynchronous development of larvae within the same cohorts”?

Response: We thank the comment by the reviewer. We replaced “synchronous” with “asynchronous” in the revised manuscript.

L332: Replace “whenever” with “whether”.

Response: We replaced “whenever” with “whether”.

Reviewer 2 Report

Overall, the study investigated gregariousness by H. vitessoides larvae. Authors report a variety of parameters in this species that are impacted by their gregarious behaviors. The novelty of the study lies in providing evidence for the impact of gregarious behavior on larval life history in this species,  and how this enhances their survival and their ecological impacts as pests. My general thoughts are below.

Given that the vast majority of the study was spent investigating the impacts of gregarious behavior on survival, developmental time, etc, and only a very small part on the transfer of avermectin, perhaps the title should be changed to reflect the content. Though the avermectin transfer is interesting and has potential ramifications, the study was very limited in this regard. For example, how much avermectin was transferred to receptors, and why does the 100 ppm treatment cause more severe impact on survivorship and emergence than the 200 ppm treatmet? There are several aspect of this study that need to be developed and the authors commented on this by stating the preliminary nature of these results.

The authors need to include significant results ( effects and interactions) in the manuscript and not direct readers to the supplementary materials.

Fig 3B could be revised or presented in a table form.

Samples were collected from October to December 2018. This is a rapid turnaround from sample collection, laboratory study and submission. Given that it takes nearly 30 days for development from egg hatch to adult, the authors need to make it clear that sampling occurred continuously.

Overall, grammar and sentence structure needs moderate revision. I recommend using editing plug-in like grammarly to this effect.

Author Response

Dear Reviewer,

We sincerely thank you for valuable comments on our manuscript. We carefully went through your comments and revised the manuscript accordingly. We are resubmitting the revised manuscript. Following is the response to your comments.

Given that the vast majority of the study was spent investigating the impacts of gregarious behavior on survival, developmental time, etc, and only a very small part on the transfer of avermectin, perhaps the title should be changed to reflect the content. Though the avermectin transfer is interesting and has potential ramifications, the study was very limited in this regard. For example, how much avermectin was transferred to receptors, and why does the 100 ppm treatment cause more severe impact on survivorship and emergence than the 200 ppm treatment? There are several aspect of this study that need to be developed and the authors commented on this by stating the preliminary nature of these results.

Response: We sincerely thank you for this valuable comment. We revised the title of the manuscript to “Larval aggregation of Heortia vitessoides (Lepidoptera: Crambidae) and evidence of horizontal transfer of avermectin”.

We provided the possible explanation why the 100 ppm treatment caused more severe impact on survivorship and emergence than the 200 ppm treatment as follows:

“Here, donor larvae exposed to the Petri dish sprayed with the high concentration (200 ppm) of avermectin solution showed reduced moving and aggregation behaviors, which might reduce the chance of contacting between donors and receptors. Therefore, donors treated with high concentration of avermectin did not cause significantly lower larval survivorship and adult emergence of receptors as compared to the controls (Fig. 7). This result showed that the high concentration of pesticide might negatively affect effectiveness of horizontal transfer, and should be avoided in H. vitessoides management.

Also. We stated that this experiment is preliminary:

It is important to note that this experiment is somewhat preliminary. Future studies are needed to investigate the factors (e.g., larval instars, donor: receptor ratio, etc.) that may influence horizontal transfer of various pesticides among gregarious larvae of H. vitessoides.

The authors need to include significant results (effects and interactions) in the manuscript and not direct readers to the supplementary materials.

Response: We included significant results (effects and interactions) in the revised manuscript as suggested by the reviewer. We also provided results of pairwise comparisons in supplementary materials.

Fig 3B could be revised or presented in a table form.

Response: We presented data in a table as suggested by the reviewer.

Samples were collected from October to December 2018. This is a rapid turnaround from sample collection, laboratory study and submission. Given that it takes nearly 30 days for development from egg hatch to adult, the authors need to make it clear that sampling occurred continuously.

Response: We provided a table (Table 1) to show that the sampling occurred continuously.

Overall, grammar and sentence structure needs moderate revision. I recommend using editing plug-in like grammarly to this effect.

Response: We used Grammarly to check the whole manuscript.